# Multimodal spatial availability: A singly-constrained measure of accessibility considering multiple modes

**Anastasia Soukhov**[1]*, **Javier Tarriño-Ortiz**[2], **Julio A. Soria-Lara**[2], **Antonio Páez**[1]

**1** Department of Earth, Environment and Society, McMaster University, Hamilton, Canada, **2** Centro de Investigación del Transporte (TRANSyT), Universidad Politécnica de Madrid, Madrid, Spain

* soukhoa@mcmaster.ca

## Abstract

Place-based accessibility measures communicate the potential interaction with opportunities at a zone that populations can access. Recent research has explored the implications of how opportunities are counted by different accessibility methods. In conventional measures, opportunities are multiply counted if more than one zone offers access to the same opportunity. This multi-count of opportunities leads to values of accessibility that are difficult to interpret. A possible solution to enhance the meaning-making of accessibility results is by constraining the calculations to match a known quantity. This ensures all zonal values sum up to a predetermined quantity (i.e., the total number of opportunities). In this way, each value can be meaningfully related to this total. A recent effort that implements this solution is spatial availability, a singly-constrained accessibility measure. In this paper, we extend spatial availability for use in the case of multiple modes or more generally, heterogeneous population segments with distinct travel behaviors. After deriving a multimodal version of spatial availability, we proceed to illustrate its features using a synthetic example. We then apply it to an empirical example of low emission zones in Madrid, Spain. We conclude with suggestions for future research and its use in evaluating policy interventions.

## Introduction

Accessibility is a key concept in the analysis of land use and transportation systems [1–3] and is coming of age from the perspective of planning research [see *inter alia* 4–8]. Beginning with the work of Hansen (1959) [1], accessibility measures have been widely used to evaluate the efficiency of transportation systems when combined with the distribution of populations and opportunities in space [9]. In this way, accessibility is a holistic measure of spatial systems' ease of reaching population-relevant destinations [10, 11].

The most common form of accessibility measure is based on the gravity model e.g. the Hansen-type measure [1]: these measures sum "weighted" opportunities around a focal point (i.e., a potential origin), based on how expensive it is to reach them. Recent research in accessibility analysis has explored the implications of methods by which opportunities are summed,

**Data Availability Statement:** All work is fully-reproducible and available within this [GitHub repository](https://github.com/soukhova/Multimodal-spatial-availability).

**Funding:** AS - Canada Graduate Scholarship - Doctoral Program (CGS D) and Michael Smith Foreign Study Supplement provided by the Social Sciences and Humanities Research Council (SSHRC). AS and PA - SSHRC's partnership grant: Mobilizing justice: towards evidence-based transportation equity policy. The funders had no role in study design, data collection and analysis, decision to publish, or preparation of the manuscript.

**Competing interests:** The authors have declared that no competing interests exist.

especially in the context of competitive accessibility [12–15]. In a typical Hansen-type accessibility measure, the sums around origins are not constrained: the same opportunity can enter the sum for different origins. Counting the same opportunity multiple times treats it as it was inexhaustible (or non-rival [16]). However, this conceptualisation may not reflect reality, as opportunities are often more exclusive than not. Some opportunities are by definition exclusive: a typical example is employment, once a job is taken up by someone in the population, the same job is no longer available for any other person to take [14, 17]. More generally, it could be conceptualised that all opportunities are somewhat exclusive so they are subject to some amount of congestion or capacity-constraint: multiple people use a specific opportunity at a given time and the more people who do, the more congested the opportunity. Congestion can be seen on a continuum, and debate on how congestion within accessibility measures can be considered is ongoing [14, 17, 18]. Nonetheless, competition has been widely considered for traditionally exclusive opportunities types such as employment and healthcare opportunities [14, 17, 19–24] and more recently for educational facilities [18, 25] as well as less exclusive amenities types such as greenspace and recreation amenities [25, 26], and shopping destinations [18, 27].

The consideration of congestion in accessibility measures was the motivation for the development of approaches that consider competition [21, 23, 24] and are popularly applied in the literature as floating catchment area methods [28]. While these approaches purport to account for congestion, Páez et al. (2019) [12] demonstrates that they do not solve the issue of multiple counting of opportunities in general, thus leading to biases in the calculation of total demand (population) and supply (of opportunities). They sometimes inflate counts and other times deflate them. In this line, recent research has paid closer attention to the way opportunities are counted in accessibility analysis. Páez et al. (2019) [12], for example, focuses on floating catchment area methods and introduces a normalization of the impedance matrix to allocate the population and then the level of service (i.e., the supply to demand) proportionally. More recently, Soukhov et al. (2023) [15] introduced a singly-constrained measure of accessibility, called spatial availability, that employs a similar but more sophisticated proportional allocation mechanism. The work of these authors show that floating catchment area methods can be seen as singly-constrained accessibility measures and improve on existing approaches by guaranteeing that each opportunity is counted only once. In other words, spatial availability treats the opportunities as *finite*. The proportional allocation mechanism of spatial availability constrains the calculations to match a known quantity, therefore ensuring that the measurements sum up to a predetermined quantity (i.e., the total number of opportunities), and so each value can be meaningfully related to this total.

A limitation of spatial availability as introduced by Soukhov et al. (2023) [15] is that it was developed for the case of a homogeneous population, for example for the case of a single mode of transportation. However, the finite nature of opportunities makes the analysis of heterogeneous populations very relevant. In the case of multiple modes of transportation, people who travel by slower modes (e.g., active modes) can usually reach fewer opportunities than people who travel by faster modes and whose range is typically far wider (e.g., car). This implies that slower travelers will often face increased competition for local opportunities from travelers who can reach said opportunities from farther afield.

This paper's primary motivation is to extend spatial availability for the case of multimodal accessibility (i.e., travel by different modes). It is worth noting though that the consideration for travel costs by different modes is in fact just one case of heterogeneous populations. The method itself can easily accommodate other forms of heterogeneity, for example: variations in travel behavior between older and younger adults (e.g., Páez et al. (2010) [29]), the propensity of older adults to use different modes of transportation (e.g., Moniruzzaman et al. (2013) [30]),

the usually shorter trip lengths of children compared to adults (e.g., Reyes et al. (2014) [31]), or the more limited travel ranges of single parents (e.g., Páez et al. (2010) [32]).

The rest of this paper is organized as follows. In Section 2, we provide a brief review of multimodal accessibility. In the following Section 3, we demonstrate the derivation of the spatial availability expression for multiple modes. In Section 4, we illustrate relevant issues through a synthetic example. This is followed in Section 5 by an empirical example using home-to-work data from the city of Madrid, Spain after the implementation of its Low Emission Zones (LEZ). Data for this example is sourced from the city's 2018 travel survey. The empirical example demonstrates the multimodal spatial availability landscape in 2018. It highlights the differences within and outside the LEZ for travelers using different modes, namely, car, transit, cycling and walking. In Section 6, we provide concluding remarks on the strengths of the use of spatial availability as a multimodal accessibility measure, and discuss potential future uses in policy planning scenarios as well as directions for future research.

## A brief review of multimodal accessibility

Place-based accessibility indicators are quantitative measures of *potential* interaction with opportunities for locations within a given region: they are summary measures of the relationship between land-use and transport systems. Arguably, the most commonly used measures are based on the gravity model [33]; this includes cumulative opportunity measures that are a special case of the gravity model [5, 10]. These gravity model based measures weight opportunities depending on the ease of reaching them. Given an origin $i$ and a destination $j$, an impedance function $f^m(c_{ij}^m)$ converts the cost of travel (e.g., time, money, generalized cost) into a score that represents the propensity for potential interaction. These measures originate from the work proposed by Hansen (1959) [1], which can take the following form in the multimodal case: $S_i^m = \sum_j O_j f^m(c_{ij}^m)$. In this form, $m$ is a set of modes that have mode-specific travel costs ($c_{ij}^m$) and/or travel impedance functions $f^m(\cdot)$.

Hansen-type accessibility is not constrained, which is to say it does not consider the opportunities as finite. To cite an example, Tahmasbi et al. (2019) [34] use Hansen-type accessibility to assess the potential interaction with retail locations by three modes: walking, public transit, and car (i.e., $m = w, p, c$). $S_i^m$ is the sum of retail locations $j$ that can potentially be reached under the travel impedance as calculated for each $i$ and $m$. In other words, for each origin $i$ three accessibility scores are calculated. In this work, Tahmasbi et al. [34] show that car travel affords the highest $S_i^m$ values in the majority of $i$ i.e., travelers who use a car can potentially reach more retail opportunities than populations using other modes. However, higher $S_i^m$ values for car do not affect the values of $S_i^m$ for other modes: in effect, each mode is analysed as if the others did not exist. Since the measure is not constrained, each opportunity is typically counted multiple times within and between modes, and as a result the sum of accessibility is not necessarily a meaningful quantity. The accessibility scores for the modes are often values that are difficult to interpret beyond making statements about relative size. For example, Lunke (2022) [35] reports accessibility scores for car in the order of tens of thousands of employment opportunities in the Oslo region. The corresponding scores for transit are lower, but still often in the thousands or tens of thousands. As reported, the ratio of the transit to the car score can be lower than 0.2 (meaning transit gives access to less than 20% of the opportunities than car). But despite the discussion about "sufficient accessibility", it is unclear what the unconstrained scores mean: is having access to 10,000 jobs by transit insufficient? After all, 10,000 employment opportunities are still plenty of opportunities. These ratios can be found elsewhere in the literature e.g., Figs 7–9 in Páez et al. (2010) [36], Fig 5 in Páez et al. (2010) [29], Páez et al. (2013) Figs 6–8 in [32]. They are useful as relative assessment of when some

members of the public are better or worse off than others, but they are silent on how bad is "worse off".

Besides ratios of accessibility, another way seen in the literature to improve interpretability of scores is to standardise them within a [0-1] range. This adjustment is only helpful insofar as it facilitates relative comparisons, but interpretation of the scores remains challenging because the values are specific to a region and convey no meaning about the magnitude of the scores. In this approach, zones always have values between 0 and 1, but how remarkable is a zone with a low score for pedestrians and a high value for car? And if remarkable, what does the difference in these standardized values mean for planners? By how much should transport systems and land-use configurations be changed to improve conditions? And in what way can these scores be used to track differences over time? Or between regions? These questions lack straightforward answers since certain values will always be relatively 'low' or 'high', but do not track to a quantity that can be intuitively understood. Presentation or discussion of Hansen-type accessibility that has been standardised in this way is not uncommon in the literature [37, 38].

If we understand opportunities to be finite and/or subject to some levels of congestion, it is possible for an accessibility measure to take on a crisper meaning. Accessibility research has a history of considering opportunity competition, especially regarding school-seats, hospital capacity, and employment opportunities [14, 15, 17–25, 39–42]. If one person reaches an opportunity—it is taken: the supply of an opportunity and the demand for that opportunity are the nodes in accessibility analysis. These types of opportunities are unambiguously exclusive. But we would go as far as to argue that every type of opportunity is subject to congestion or capacity constraint, even when the opportunities are conventionally seen as inexhaustible.

Amenities are a good example of this. For instance, standards for providing green spaces are often stated in the form of *exclusive access*, in units of amenity per capita. For example, a 2013 planning document for the Ile-de-France region suggested a public green space municipal standard of 10 $m^2 per inhabitant$ [43]. Green spaces are not evenly distributed, meaning those who have access to them depends on where they are and how easy they are to reach. This formulation of amenity provision is not unusual. As another example, Natural England recommends a national "accessible natural greenspace standard" such that the minimum supply of space is 1 $ha$ of statutory local nature reserves per thousand population [44]. Similarly, the World Health Organization [45] recommends that cities provide a minimum of 9 $m^2$ of green area per inhabitant. For our purposes, standards of this type translate into "how much of this resource is available to one individual that has not been claimed by anyone else?". Green spaces often have large capacities, but they still have a capacity and it is not the same for a person to have access to 5 $m^2$ of *uncongested* green space as 15 $m^2$. This difference is in fact a matter of justice [43, 46]. Constraining accessibility is a useful way to evaluate the congested availability of any type of opportunity. As development of sound standards is emphasized in the planning literature, in particular in regards to fairness in transportation [47], spatial availability analysis can be used to develop and assess standards.

The relevance of the considerations above is put in sharper relief when we think about the use of multiple modes (or heterogeneous populations). If we return to Oslo for a moment [35], we notice that the places that have high accessibility by transit are also the places that have *very high* accessibility by car (in their Fig 2). Those two populations are going for the same opportunities, and those travelling by transit have fewer to choose from the start. More generally, people in a zone who are advantaged with relatively low cost of travel will have the ability to potentially reach more opportunities than other people. Due to this advantage, through the perspective of finite opportunities, there are fewer opportunities left for everyone else, especially for those who use modes that are slower or otherwise more expensive.

As noted in the Introduction, competitive accessibility was the rationale for developing floating catchment area methods (FCA), popularized by Luo et al. (2003) [28] who reformulated the work of Shen (1998) [24] into two steps (although similar, and earlier, developments are found in [21, 23]). Shen-type accessibility is formulated as: $a_i^m = \sum_j \frac{O_j f^m(c_{ij}^m)}{\sum_m D_j^m}$ where $D_j^m$ is the potential demand for opportunities equal to travel impedance weighted population $\sum_i P_i^m f^m(c_{ij}^m)$ and the remaining variables are repeated in the Hansen-type measure. Shen-type modal accessibility ($a_i^m$) can be understood as a ratio of the travel impedance-weighted supply of opportunities for $m$-mode in $i$ over the travel impedance-weighted demand for opportunities. In this way, it considers competition. That said, the measure remains unconstrained, meaning both population *and* opportunities are multiply counted [12]. In other words, interpretation of the Shen-type accessibility scores between modes is fraught, as it is for Hansen-type measures.

To illustrate, Tao et al. (2020) [48] calculates $a_i^m$ to jobs for different income-group populations in Shenzhen, China for those using transit and car. Their results indicate that zones with low-income populations have lower $a_i^m$ than zones with higher-income populations. Further, they show that $a_i^{\text{transit}}$ is lower than $a_i^{\text{car}}$ in many zones, arguing that this may further place those zones with lower-income populations at a disadvantage. $a_i$ and/or $a_i^m$ are used to compare relative spatial differences in overall competitive accessibility and multimodal competitive accessibility, but because opportunities were doubly counted (entering the sums of both modes), this makes for uneasy interpretations of the differences in $a_i^m$ between modes. Questions that this approach leaves unaddressed include: what is the impact of competition on the difference in $a_i^m$ values? How does the impact vary spatially? And what is the interpretation of this difference?

Spatial availability improves on the discussed Hansen-type and Shen-type accessibility approaches by constraining the sum of opportunities, that is, by treating opportunities as finite. This is done by means of proportional allocation factors that follow well established principles of spatial interaction and the gravity model [49]. In Soukhov et al. (2023) [15] these factors consider: the mass effect (e.g., the size of populations at different origins); and the cost of travel from different zones (e.g., some sub-populations face relatively higher or lower costs). The following section introduces the multimodal form of spatial availability.

## Multimodal spatial availability

In brief, we define the spatial availability $V_i$ at an origin as the proportion of all opportunities in the region that are allocated to origin $i$ from all destinations $j$. The general formulation of spatial availability is shown in Eq (1) [15]:

$$V_i = \sum_{j=1}^{J} O_j F_{ij}^t \tag{1}$$

Where:

- $F_{ij}^t$ is a balancing factor that depends on the size of the populations at different locations that demand opportunities $O_j$, as well as the cost of movement in the system $f(c_{ij})$.

- $V_i$ is the number of spatially available opportunities at $i$; the sum of $V_i$ is identical to the total number of opportunities in the region (i.e., $\sum_j O_j = \sum_i V_i$).

Compared to Hansen-type accessibility $S_i = \sum_{j=1}^{J} O_j f(c_{ij})$, we see that spatial availability is, like the Hansen-type measure, a weighted sum of the opportunities. What makes spatial

availability stand apart from other approaches is how the weight used in the sum (balancing factor $F_{ij}^t$) implements a proportional allocation mechanism to ensure that the sum of $V_i$ is constrained to match the total number of opportunities in the region. As such, spatial availability is singly-constrained and natively considers competition. $F_{ij}^t$ consists of two parts. The first part is a population-based proportional allocation factor to model the mass effect of the gravity model:

$$F_i^p = \frac{P_i}{\sum_i P_i}$$

This factor makes opportunities available based on demand. Secondly, there is an impedance-based proportional allocation factor that models the cost effect:

$$F_{ij}^c = \frac{F_{ij}^c}{\sum_j F_{ij}^c}$$

This factor makes opportunities available preferentially to those who can reach them at a lower cost. $F_i^p$ and $F_{ij}^c$ are designed so that they both equal 1 when summed across all $i$ in the region (e.g., $\sum_i F_i^p = 1$ and $\sum_i F_{ij}^c = 1$). These factors are combined multiplicatively to yield $F_{ij}^t$ which ensures that a proportion of the opportunities $O_j$ are allocated to each $i$ accordingly. In other words, assuming a finite number of opportunities in the region, $F_{ij}^t$ proportionally allocates $O_j$ to each $i$ such that the resulting $V_i$ value represents the number of opportunities *available* to the population at $i$. Each zonal value is a proportion of the opportunities in the region (i.e., $\sum_j O_j = \sum_i V_i$).

The focus of this paper is to extend $V_i$ for the measurement of multimodal applications (or more generally heterogeneous populations). To do so, the balancing factors are reformulated so that 1) the mass effect now accounts not only for the size of the population at $i$, but also the size of sub-populations within $i$; and 2) the cost of travel is not only for different zones, but by sub-populations within each zone (e.g., the cost of travel from $i$ by car, transit, walking, etc.) relative to all zones. When we introduce modes (or sub-populations) $m$, the proportional allocation factors need to satisfy the condition that $F_i^{pm}$ and $F_{ij}^{cm}$ can be summed across each $m$ at each $i$ and then across all $i$ to equal to 1. They are also similarly combined multiplicatively to obtain their joint effect, represented as the combined balancing factor $F_{ij}^{tm}$ similar to that detailed in Eq (2). This factor is given by:

$$F_{ij}^{tm} = \frac{F_i^{pm} \cdot F_{ij}^{cm}}{\sum_{m=1}^{M} \sum_{i=1}^{N} F_i^{pm} \cdot F_{ij}^{cm}} \tag{2}$$

where:

- The factor for allocation by population for each $m$ at each $i$ is $F_i^{pm} = \frac{P_i^m}{\sum_m \sum_i P_i^m}$; and

- The factor for allocation by cost of travel for each $m$ at $i$ is $F_{ij}^{cm} = \frac{f^m(c_{ij}^m)}{\sum_m \sum_i f^m(c_{ij}^m)}$

Implementing $F_{ij}^{tm}$, Eq (3) gives the multimodal version of spatial availability $V_i^m$:

$$V_i^m = \sum_{j=1}^{J} O_j \ F_{ij}^{tm} \tag{3}$$

Where:

- $m = 1, 2, \cdots, M$ is a set of $M$ modes (or sub-populations) of interest.

- $F_{ij}^{tm}$ is a balancing factor $F_{ij}^{t}$ for each $m$ at each $i$.

- $V_i^m$ is the spatial availability $V_i$ for each $m$ at each $i$; the sum of $V_i^m$ for all $m$ at each $i$ is equivalent to the total sum of opportunities in the region (i.e., $\sum_j O_j = \sum_i V_i = \sum_m \sum_i V_i^m$).

Next we use a synthetic example to contrast multimodal spatial availability with multimodal versions of Hansen-type accessibility (unconstrained) and Shen-type (unconstrained and competitive) accessibility.

## An illustrative synthetic example

Consider the simple system shown in Fig 1. The figure shows a region with population at three population centers ($A$, $B$, $C$) and jobs at three employment centers (1, 2, 3). The population at each origin $i$ is consists of two sub-populations, one using a faster mode $z$ and another using a slower mode $x$, to travel to employment centers. Population center $A$ is Suburban: it is closest to its own relatively large employment center at 1, close to the Urban's equally large employment center 2, and has a population that is smaller than the Urban $B$ and larger than the Satellite $C$. $B$ has the largest $x$-using population (40%), followed by $A$ (33%) then $C$ (30%). This synthetic example is inspired by the single-mode example used in Shen (1998) [24] and reconfigured in Soukhov et al. (2023) [15].

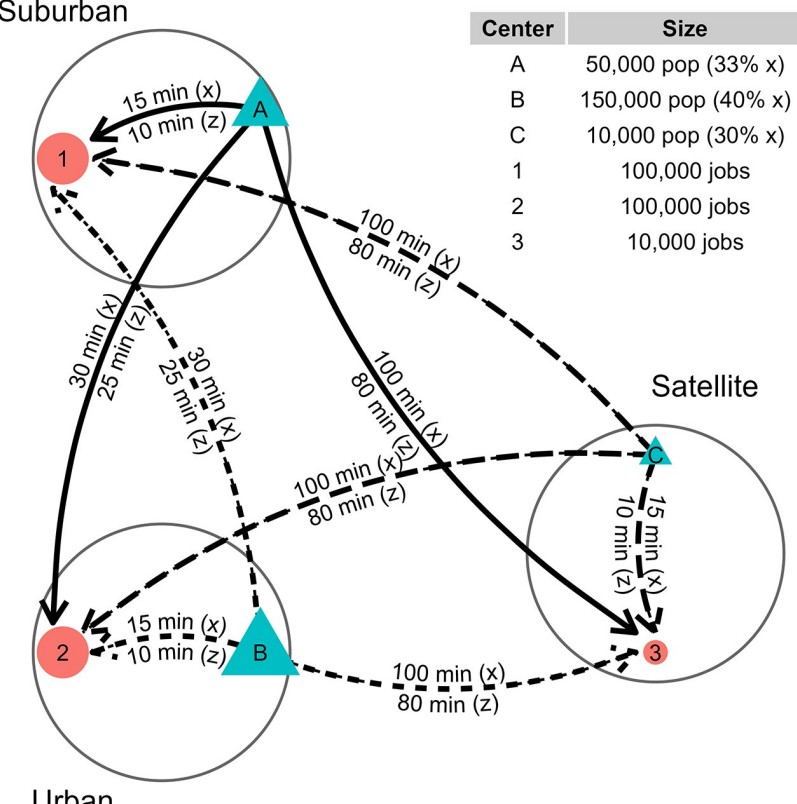

| Center | Size |
|---|---|
| A | 50,000 pop (33% x) |
| B | 150,000 pop (40% x) |
| C | 10,000 pop (30% x) |
| 1 | 100,000 jobs |
| 2 | 100,000 jobs |
| 3 | 10,000 jobs |

**Fig 1. Multimodal synthetic example: Locations of employment centers (in orange), population centers (in blue), number of jobs and population, and travel times for two modes (slower mode x and faster mode z).**

From the perspective of access to a *finite* amount of opportunities in the region (210, 000 jobs), the sub-population that is most proximate to jobs (lowest cost to reach), furthest from large populations (least competition), and uses the fastest mode $z$ (greatest range) can potentially reach the largest number of opportunities. This appears to be the sub-population at $A$ using mode $z$. Sub-populations located in opposite conditions (i.e., more distant from jobs, close to large populations, and using slower mode $x$) are at a relative disadvantage. The competition for opportunities between different mode-using populations matters as it reflects how well the land-use and transport system serves (or does not serve) certain populations.

The values calculated for $S_i^m$ (Hansen-type accessibility), $a_i^m$ (Shen-type accessibility), and $V_i^m$ (spatial availability) for each $i$ and $m$ are shown in the middle three columns and are aggregated for each $i$ in the final two columns in Table 1. As in the example in Shen (1998) [24], we use a negative exponential impedance function $f^m(c_{ij}^m) = \exp(-\beta \cdot c_{ij})$ with $\beta = 0.1$ for both $x$ and $z$ modes for all accessibility measures calculations. Notice that in this example we use the same impedance function but the travel times are different for the two modes. More generally, it is possible to use different impedance functions for the modes, as the empirical example in the following section demonstrates.

Hansen-type accessibility $S_i^m$ is presented for each origin and mode in the third column of Table 1. For all $i$, the travel by $z$ results in higher values of $S_i^m$ than travel by $x$. Lack of competition, or alternatively the assumption of an inexhaustible resource in the calculation of $S_i^m$, leds to a curious result. Since the populations in $A$ and $B$ have the same travel impedance to employment centers 1, 2 and 3 (either 15, 30, or 100 minutes using $x$ or 10, 25, or 80 minutes using $z$), their values of $S_i^m$ are the same for both $A$ and $B$. Furthermore, the total sum of $S_i^m$ in the region is equal to 150,570.2. This value lacks an intuitive interpretation: it represents the weighted sum of opportunities that may be reached within the region according to the travel impedance (i.e., the travel behavior and the characteristics of the modes) and does not usefully translate into any sort of benchmark. To connect this example to the aforementioned literature, $S_i^m$ is calculated in the work of Tahmasbi et al. (2019) [34]; they contrast differences in $S_i^m$ values between modes in a relative and comparative sense, but make no further interpretation of the $S_i^m$ values. More densely populated metropolitan regions will tend to have more opportunities and hence large $S_i^m$ values and less densely regions, smaller values; how much of these differences may simply an artifact of region density?

In the fourth and sixth columns in Table 1 the results for Shen-type accessibility are reported: first for both origin and mode $a_i^m$ as well as aggregated by the weighted mean mode-population ($\sum_m \frac{P_i^m}{P_i} * a_i^m$) to represent a value for each origin $a_i$. Unlike $S_i^m$, this measure does considers *competition*. For instance, the population travelling by $x$ from $A$ and $B$ do not have the same values of $a_i^m$ as those travelling by $z$. In fact, $A$ has the highest values $a_i^m$ and $a_i$ values

**Table 1. Accessibility values (S, a, V) at each origin (i) for the synthetic example.** Displayed per mode (m) (columns three to five) and aggregated per i (columns six and seven).

| i | m | $S_i^m$ | $a_i^m$ | $V_i^m$ | $a_i$ | $V_i$ |
|---|---|---|---|---|---|---|
| A | x | 27,292.18 | 0.95 | 15,696.89 | 1.36 | 67,482.61 |
|   | z | 44,999.80 | 1.57 | 51,785.72 |  |  |
| B | x | 27,292.18 | 0.64 | 38,170.03 | 0.88 | 132,638.94 |
|   | z | 44,999.80 | 1.05 | 94,468.91 |  |  |
| C | x | 2,240.38 | 0.68 | 2,035.86 | 0.99 | 9,878.45 |
|   | z | 3,745.89 | 1.12 | 7,842.59 |  |  |
| TOTALS |  | 150,570.22 | N/A | 210,000.00 | N/A | 210,000.00 |

since this center has the lowest travel impedance to opportunities (lower than at $C$, $A$ and $B$ are equal) and faces relatively low competition, not being close to a relatively large population (lower than at $B$).

However, the calculations of $a_i^m$ are not constrained: the total sum of $a_i^m$ or $a_i$ is practically meaningless since it represents a sum of ratios. For instance, the population travelling by $z$ from $A$ has a value of 1.57 jobs per job-seeking population compared to 0.95 for users of mode $x$. What is the meaning of these values? The difference between these modes is equal to 0.62, but 0.62 of what? How many more job opportunities can users of $z$ reach compared to user of $x$? When $a_i^m$ is aggregated to $a_i$ as shown in the sixth column, the values face similar interpretability issues. The Shen-type measure is implemented in aforementioned work of Tao et al. (2020) [48] to calculate modal $a_i^m$ values and the aggregated $a_i$ is implemented in the work of Carpentier et al. (2020) [50]. However, similar to Hansen-type accessibility, these works discuss relative and spatially comparative differences in values, but veer from interpreting the values of $a_i^m$ or $a_i$ themselves. In fairness, interpretation is complicated by the multiple counting of opportunities between zones and modes.

In contrast, spatial availability $V_i$ considers competition and is constrained such that the total sum of values is equal to the total number of opportunities in the region (i.e., 210, 000 jobs). Seen in fifth column of Table 1, the values of $V_i^m$ for $A$ and $B$ are not the same within each mode (as this measure considers competition). In fact, at $A$, users of mode $z$ capture 36,088.84 more spatially available jobs (of the 210, 000 jobs in the region) than the sub-population travelling by the slower-mode $x$. The numerical difference is clear since it refers to opportunities out of the total.

Furthermore, the proportional allocation mechanism also means that the values of $V_i^m$ for any origin $i$ can be aggregated across $m$ and compared between zones ($V_i = \sum_m \sum_i V_i^m$). This aggregation, $V_i$, is shown in the seventh column in Table 1. $A$ is allocated 67,482.61 spatially available opportunities for both modes. 77% of this spatial availability allocated to $A$ is assigned to users of mode $z$ despite representing 66% of $A$'s population.

Spatial availability can be further aggregated to better interpret competition between modes. Across the entire region, 130,000 people use $z$ (62% of the region population). However, users of $z$ account for 73% of the region's total spatial availability—while the remaining 27% is allocated to users of mode $x$ who are 38% of the total population. Notably, the population who uses $x$ have 11% fewer spatially available opportunities than its share in the population. This realization could lead us to ask normative questions: how unequal should availability of opportunities be by mode? What intervention could help to redistribute spatial availability to sub-populations commensurate with their proportion of the total?

Since spatial availability is constrained to the total opportunities in the region, the values at $i$ have a straighforward interpretation. Inequality in $V_i^m$ values can be explored through a variety of approaches. For instance, consider travel times. The population of travelers who use $z$ accounts for 67% of the potential travel time traveled in the region: this is 7% less travel time than the proportion of spatial available opportunities that is allocated to them. In other words, the population of users of $z$ travels fewer minutes overall and has more spatial availability of opportunities than users of the slower mode $x$.

Alternatively, inequities in spatial availability between modes can be explored through proportional benchmarks. A spatial availability per capita $v_i^m$ is presented in Eq (4):

$$v_i^m = \frac{V_i^m}{P_i^m} \tag{4}$$

The values of $v_i^m$ for $A$, $B$, and $C$ for users of $x$ are 0.95, 0.64 and 0.68 spatially available jobs per capita, respectively. The values of $v_i^m$ for users of $z$ are much higher, with values of: 1.57, 1.05 and 1.12 respectively. Users of $x$, especially those at $B$ and $C$, are directly impacted by the jobs that are spatially available to users of $z$ *in addition to* the mass effect (occurring at $B$, high population density) and high travel impedance (occurring at the Satellite $C$). Notably, $v_i^m$ values are equal to the ratios of the Shen-type measure $a_i^m$ in this case, however they take on a different interpretation.

If, let us say, the planning goal was to have one spatially available job per mode-using population, a policy intervention could be devised, to reduce the values of $v_i^z$ (making it slower or more expensive) and increase the values of $v_i^x$ (making it faster or less expensive). The purpose of this synthetic demonstration is to show how spatial availability can be used to quantify the competitive (dis)advantage in a multimodal application. In what follows, we demonstrate the use of multimodal spatial availability through an empirical example.

## Empirical example

### Context

The context for the empirical example is the city of Madrid, Spain. This city implemented a Low Emission Zone (LEZ) in 2017 to pursue goals set out in the national climate change agenda such as cutting nitrogen dioxide levels and prioritizing people's movement in the city. LEZs elsewhere have similarly been implemented as interventions to reduce GHG emissions, improve air quality, and support sustainable mobility [51, 52]. Though the rules of exclusion vary by city, LEZs aim to deter/reduce traffic in designated zones under threat of penalty (e.g., fines, seizure of vehicle). In other words, LEZs implement a form of *geographic discrimination* as they change how people can reach opportunities by making it more costly for some forms of travel, typically cars, to circulate in predetermined zones. When considering opportunities as finite in a region, this discrimination reduces the competition of one mode and opens up opportunities for other modes to better thrive. At their core, LEZs operate by changing the accessibility landscape of a city from the perspective of multiple modes.

In geographic scope, the 2017 boundaries of the LEZ in Madrid were relatively modest, covering only approximately 4.72 km$^2$ of the central business district of the city (the so-called LEZ Centro). As of this writing, there are plans to expand these boundaries to the area inside the M-30, an orbital highway in proximity to the city center. Within the 2017 LEZ Centro implementation, all cars, motorcycles and freight vehicles with environmental labels A or B (older makes and models of fossil fuel internal combustion engine vehicles), were disallowed from entering the zone unless they are used by residents or meet other exemptions. This restriction impacted approximately half of all car trips that used to travel into what is now the LEZ Centro [53].

The purpose of this empirical example is to quantify spatial availability to employment opportunities by different modes in Madrid after LEZ Centro implementation. Particularly, we demonstrate how $V_i^m$ can be used to illustrate the spatial availability advantage that more sustainable modes (that are often slower than car) gain within/around the LEZ Centro. We speculate on how this competitive advantage is heightened as a result of the LEZ implementation that restricts car mobility.

### Data

The source of origin-destination data for our empirical example is the 2018 Travel Survey of the Community of Madrid [54]. This is a representative survey that offers a snap-shot of travel

patterns for a typical weekday in 2018. The survey collected 222,744 trips from a representative sample of 85,064 households across the traffic analysis zones (TAZ) in the Community of Madrid. For context, the population older than 3 years in the Community is 6,507,184.

In this example, we use all direct home-to-full-time-work trips, by all modes. The trips are expanded using population weights. Figs 2 and 3 show the number of workers and the distribution of full-time jobs in the City of Madrid by TAZ. The TAZ shapefiles are available from the Community of Madrid open data portal [55]. The pink boundary represents the LEZ Centro in effect in 2017 and reflected in the 2018 travel survey. The purple boundary represents the LEZ planned for the boundaries of the M-30 highway and is present in the plots as a spatial reference for areas in proximity to the LEZ Centro.

The total sum of jobs $O_j$ are shown in Fig 2 and the populations that go to a work destination by four modal categories $P_i^m$, is displayed in Fig 3. The modal shares in Fig 3 are calculated based on the primary mode specified in the survey and summarized into four categories as follows:

- Car/motor: all cars and operating modes (e.g., cab, private driver, company, rental car, main driver of a private car, passenger in a private car) and all public, private or company motorcycle/mopeds.

- Transit: all bus, trams, and trains.

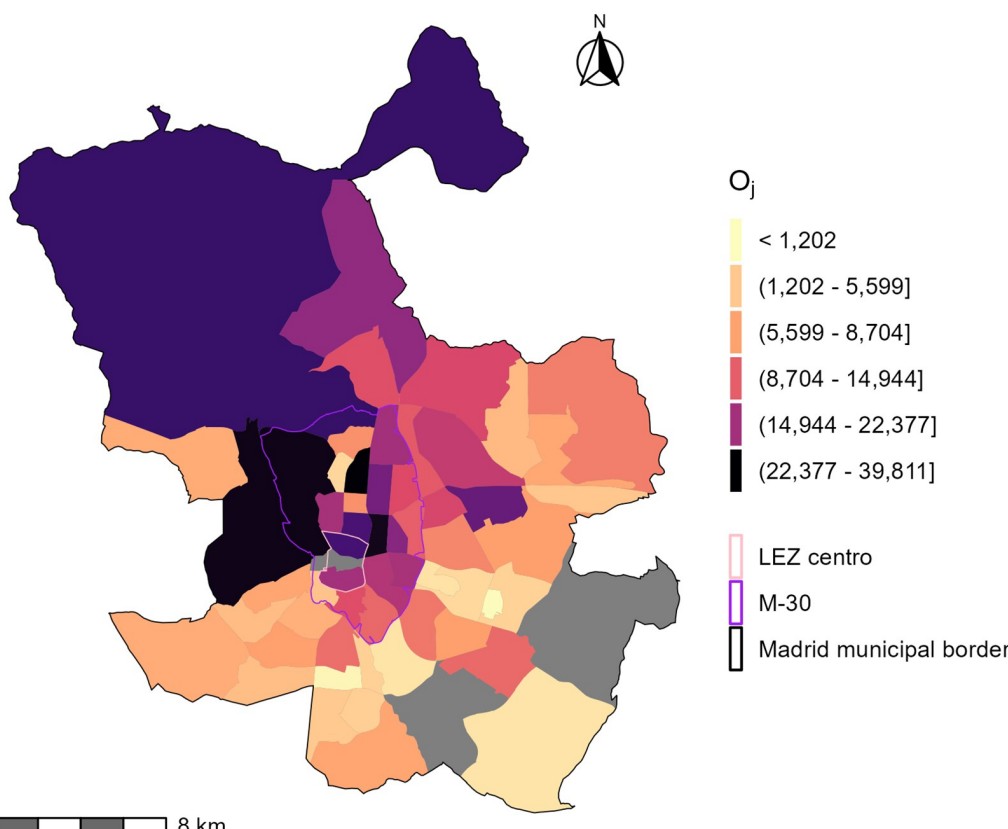

**Fig 2. Distribution of jobs taken by people living and working in Madrid as reported in the 2018 travel survey.** Grey TAZs have no jobs. Ranges of values in the legend are quintiles. The TAZ shapefile is available from the Community of Madrid open data portal.

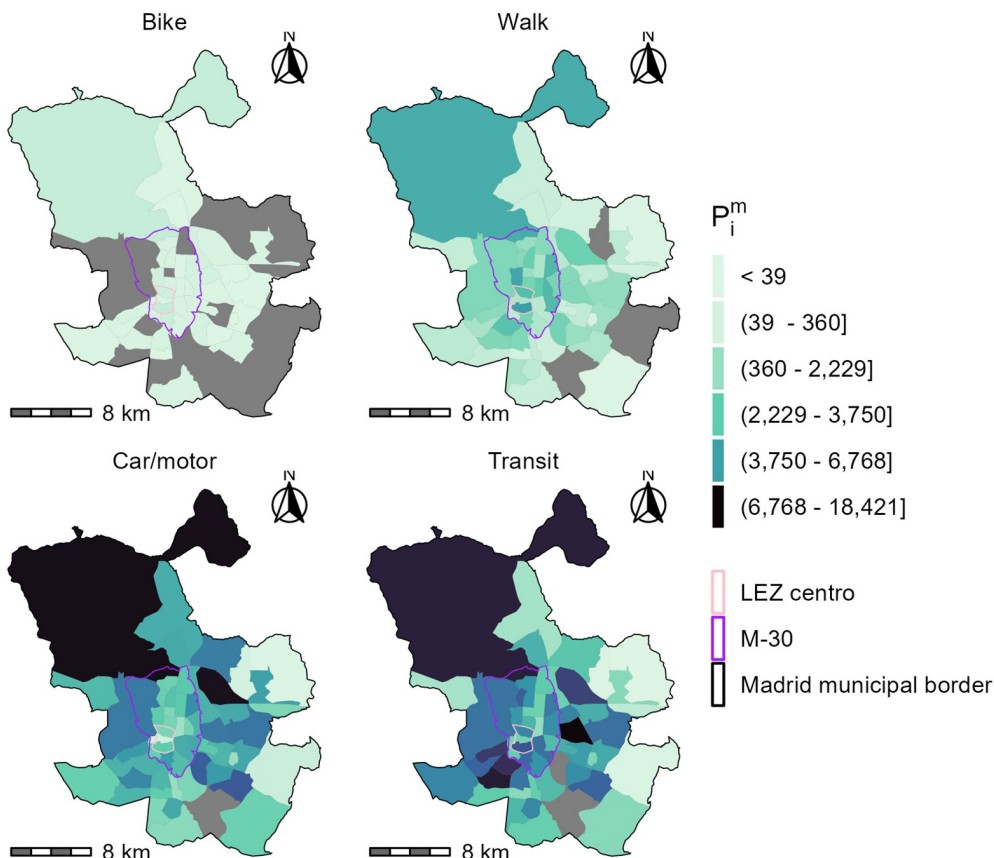

**Fig 3. Population living and working in Madrid by mode of transportation as reported in the 2018 travel survey and represented at the level of TAZ.** Grey TAZ have no population. Ranges of values in the legend are quintiles. The TAZ shapefile is available from the Community of Madrid open data portal.

- Bike: all bicycle trips (e.g., private, public, or company bike trips) and "other" types of micro-mobility options.

- Walk: pedestrian mode.

Some aggregation of modes is necessary to calculate the travel impedance functions by mode. From Fig 2, the largest concentration of jobs is within, near, and to the north of LEZ Centro. The populations with access to those jobs by mode (Fig 3) are spatially distinct. Travel by car and transit represent 37% and 47% of the modal share respectively. The population that travels by transit is more spatially distributed than those using cars—particularly near and within LEZ Centro. This distribution is likely caused by a variety of factors including: transit coverage and service within with city, effective car infrastructure outside of the M-30, and/or the impact of the LEZ Centro itself. From Fig 3, active travel is less common than motorised trips at 1% and 15% for cycling and walking respectively. Noticeably, there is a positive trend between the walking and cycling in zones where transit is also present. This positive trend is higher than for car-using populations.

Travel times are provided within the travel survey by mode. This information is used to calibrate mode-specific travel impedance functions $f^m(c_{ij}^m)$. To illustrate the modal differences in travel times, the following descriptive statistics per mode are presented:

- Car/motor: mean 36 minutes (min: 0 minutes, Q2: 15 minutes, Q3: 55 minutes, max: 120 minutes)

- Transit: mean 55 minutes (min: 1 minutes, Q2: 30 minutes, Q3: 80 minutes, max: 120 minutes)

- Bike: mean 34 minutes (min: 5 minutes, Q2: 15 minutes, Q3: 40 minutes, max: 115 minutes)

- Walk: mean 27 minutes (min: 1 minutes, Q2: 10 minutes, Q3: 45 minutes, max: 119 minutes)

Impedance functions $f^m(c_{ij}^m)$ are calibrated from the travel times in the survey via the empirical trip length distribution (TLD). An empirical TLD is given by the proportion of trips at various travel cost bins. This distribution is then used to estimate the parameters of a function for the travel impedance (as done in [56–58]). To fit the impedance functions, we use the Maximum likelihood estimation and the Nelder-Mead method for direct optimization available within the {fitdistrplus} R package [59]. Based on goodness-of-fit criteria and associated diagnostics, the gamma and log-normal probability density functions are selected as best fitting curves for the motorised and non-motorised modes respectively. The selection of functional forms aligns with empirical examples in other regions [15, 60, 61]. The shape and rate parameters for the gamma functions (motorised modes) are 1.8651852 and 0.051468 for car/motor and 2.7566235 and 0.0499193 for transit; for the log-normal functions (non-motorised modes), the mean and standard deviation parameters are 3.2372212 and 0.7575986 for bike and 2.9918042 and 0.7575986 for walk.

Fig 4 includes four plots to visualize the calibrated impedance functions (represented as black lines) superimposed on the empirical TLD. The impedance functions can be interpreted as the propensity to travel (y-axis) given a trip travel time (x-axis). The functions reflect a combination of possibilities and preferences: the travel behavior given the transportation technologies available. For example, trips shorter than 5 minutes do not occur frequently for any mode; this reflects the spatial separation between places of residence and places of work commonly seen in many cities. In terms of the non-motorised modes, there is a preference towards walking trips around 15 minutes in duration, as seen from the highest value of $f^{walk}(c_{ij}^{walk})$. With respect to travel by bicycle, longer travel times are more common; although the highest value of the impedance also corresponds to approximately 15 minutes, the curve has a longer tail and values decrease less rapidly at longer travel times than is the case of $f^{walk}(c_{ij}^{walk})$. A similar trend can be observed for the motorised modal options where transit mode is more spread out than car/motor mode. All in all, these functions represent the propensity of travel by mode by duration of trip, and are used to calculate the proportional allocation factors $F_{ij}^m$ for $V_i^m$.

## Results

Fig 5 illustrates the multimodal spatial availability landscape for each of the four modes $m$ at the level of traffic analysis zones $i$. $V_i^m$ is a proportion of the total number of 847,574 jobs in the city. Since $V_i^m$ is calculated based on the population of workers and the distribution of jobs, the values can be understood as the number of full-time jobs that are spatially available to full-time workers at that $i$ traveling by mode $m$, relative to all the jobs in the city.

Fig 5 is a snapshot of the spatial availability as reflected by the multimodal origin-destination flows and travel times from the 2018 travel survey: it incorporates the travel behaviour after LEZ implementation i.e., reduced car trips into the LEZ Centro. Furthermore, this implementation of spatial availability assumes that all employment opportunities are of interest to the entire population. Also, opportunities are proportionally allocated to mode-using

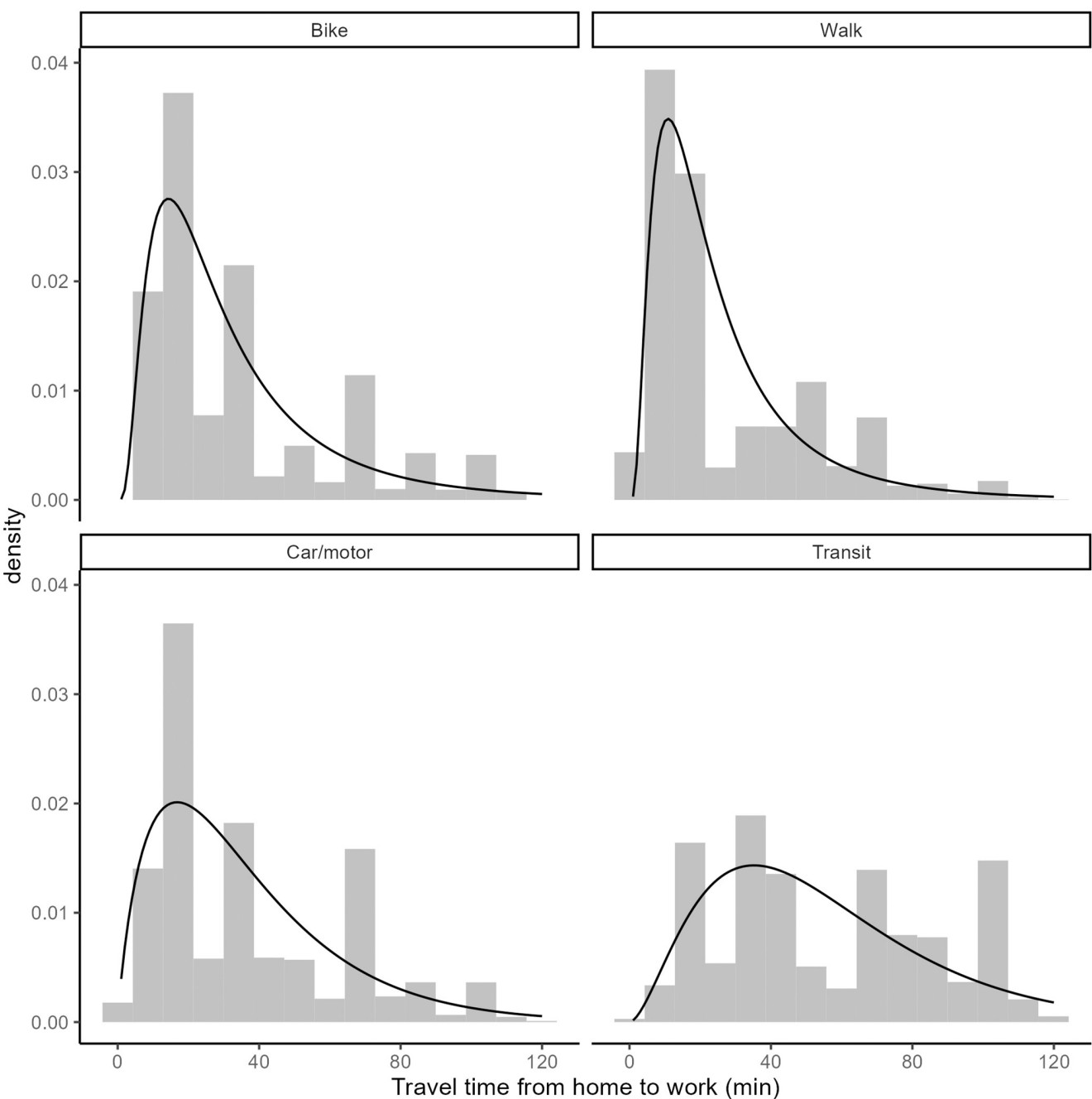

**Fig 4. Fitted impedance function against empirical TLD (bars) corresponding to the home to full-time work origin destination flows for the city of Madrid from the 2018 travel survey.**

populations based on travel time and mode-using population residing in the zones relative to the total travel time and population.

There are noticeable differences in the magnitude of $V_i^m$ between modes. In Fig 5, the majority of $V_i^m$ are allocated to workers travelling by car and transit. This can be expected since users of these modes represent 84.1% of the total population. The ability to travel at greater speeds also advantages these modes compared to the non-motorized modes. However,

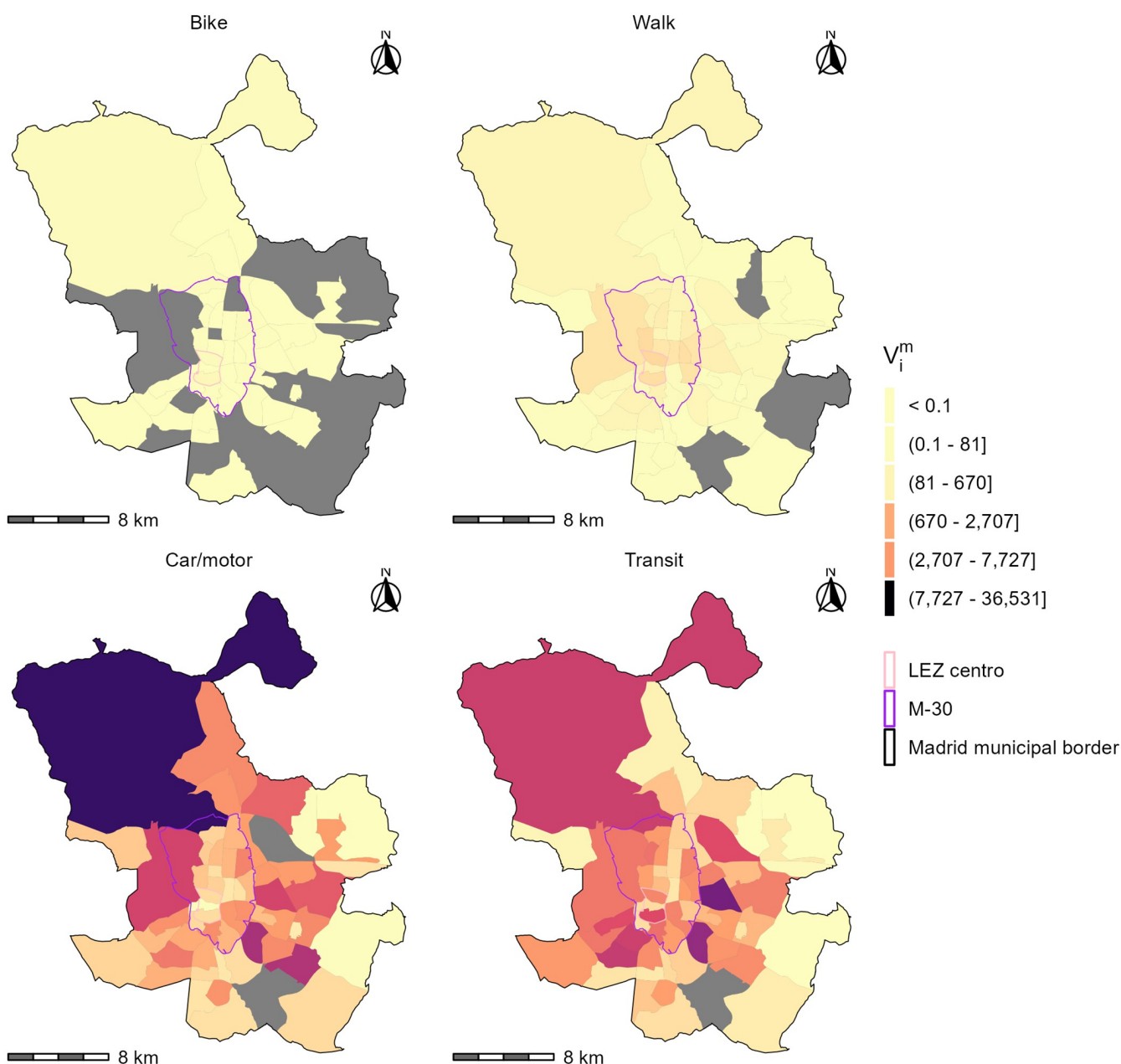

**Fig 5. Spatial availability of jobs per origin and mode $V_i^m$ in Madrid at the level of TAZ.** Grey TAZ have no population. Ranges of values in the legend are quintiles. The TAZ shapefile is available from the Community of Madrid open data portal.

differences in $V_i^{\mathrm{car}}$ and $V_i^{\mathrm{transit}}$ values exist in space: car users outside of the M-30 are allocated greater spatial availability, while some zones inside the M-30 have greater spatial availability for transit. Overall, the magnitude of $V_i^m$ values for non-motorized mode-users are lower than for car and transit, but the highest values of $V_i^{bike}$ and $V_i^{walk}$ tend to be found in zones within the M-30 and origins with higher $V_i^{transit}$.

To highlight the spatial differences in modal competitive advantage, Fig 6 displays the spatial availability and population per mode aggregated for three areas of the city. Shifting focus

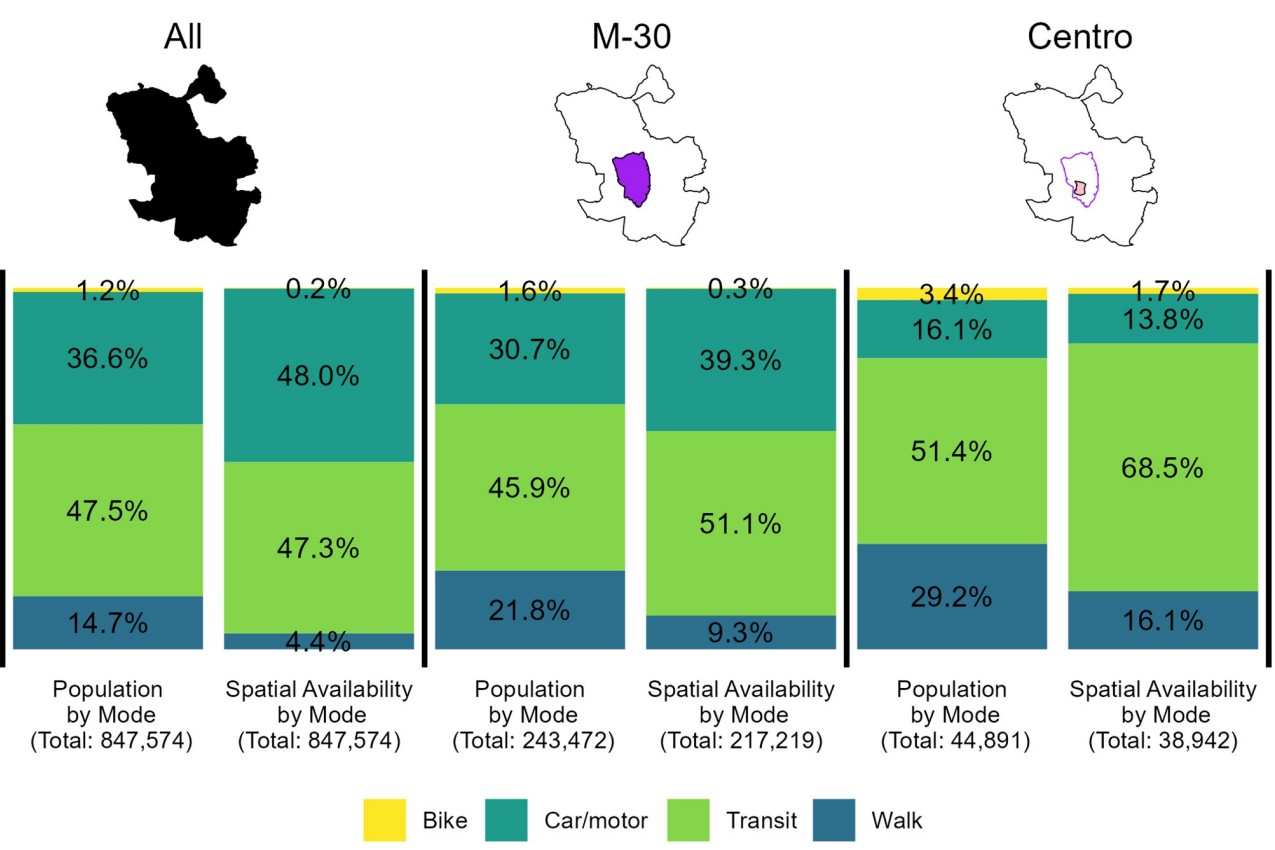

**Fig 6. Proportion of population by mode and spatial availability of jobs by mode aggregated for three areas.** From left to right, the city of Madrid (All), the area within the M-30 highway (M-30), the area within the Centro region (Centro).

to the left-most bars of Fig 6, motorised mode users can avail 95.3% of all jobs in the city (Spatial Availability by Mode). However, car users are allocated a disproportionate share of spatial availability relative to their city-wide population. The combined population of car and transit users is 36.6% and 47.5% respectively, and these populations are respectively allocated 48.0% and 47.3% of the city's job availability. When conceptualising the number of opportunities accessed as a finite value (total: 847,574 jobs), fewer opportunities are spatially available to lesser competitive modes. Modes are at a disadvantage when relative travel cost is high (see Fig 4) and the mode-using population is relatively small compared to all populations (especially populations with lower travel costs).

Though car mode offers the most spatial availability overall, this is not the case within the Centro. As summarized in the two right-most bars in Fig 6, the proportion of jobs spatially available to car users in Centro is 13.8% (5,373 opportunities), less than the proportion of the car users in the Centro (16.1%). The trend in the Centro for car-users is opposite to that of the city overall (left-most bars) and the areas inside the M-30 (middle bars). We suspect that car-mode's competitive advantage is blunted by the LEZ: the number of car trips are relatively reduced in the area making non-car modes more competitive. Car mode's advantage in the Centro is also diminished by the relative increment of the mass effect and concentration of jobs (Fig 2) within/around the Centro.

Since car mode is less competitive within the Centro, other modes are relatively *more* competitive. Referring to the active travel modes in the right-most bars in Fig 6, 1.7% and 16.1% of

opportunities are spatially available to bike and walk modes respectively, while their populations represent 1.2% and 14.7% of the population. The disparity between the proportions of cyclists and walkers and the proportions of jobs spatially available to them is *smaller* than displayed in the other two aggregations (All and M-30). We suspect that by restricting the ability of cars to enter Centro, the LEZ contributes to leveling the playing field for slower modes, in particular cycling and walking but also transit. Transit users are generally close to parity city-wide (left-most bars), with nearly as many spatially available jobs as transit users. Still, transit mode has the greatest advantage in the Centro with 68.5% of spatially available jobs for 51.4% of transit users in Centro. This result makes intuitive sense: after car, transit is the mode with the greatest range and, unlike car, it faced no restrictions by the LEZ.

The spatial differences in the competitive (dis)advantage of spatial availability between modes can also be visualized at a finer level of spatial granularity. Fig 7 shows $v_i^m$, the spatial availability $V_i^m$ divided by the population of users of $m$. Values of $v_i^m$ below one are shown in shades of orange, and indicate TAZs with less than one spatially available opportunity per capita for the mode. Values above one are shown in shades of green, and indicate TAZs with more than one spatially available opportunity per capita for the mode. The highest spatial availability per capita (shown in blue) is for car users in a zone northeast just beyond the M-30. These plots illustrate in unambiguous fashion, and in a quantity that is comparable over space and time, the advantage in terms of spatial availability of car for most of the city (bottom left plot, areas denoted with green $v_i^m$ values above one). It can also be observed that spatial availability of jobs is relatively well balanced for transit users over most of the regions (i.e., many zones are light green). In contrast, spatial availability of jobs for non-motorised modes is low (under one) overall, although less so within/around the LEZ Centro.

Since $v_i^m$ values are comparable across regions and over time, Fig 7 potentially provides a benchmark for quantifying changes in LEZ policies in the future. Even within the M-30, $v_i^c ar$ values are still high (over 1) for most zones. These results may give reasonable grounds to speculate that a spatial expansion of the LEZ to include all areas within the M-30 would increase the spatial availability of jobs for transit users, cyclists and pedestrians. However, further investigation is needed.

## Discussion and conclusions

Accessibility measures are an important tool in transportation research [9] and are increasingly seen as valuable for planning purposes [4–8]. They boast a long history of development, beginning with Hansen-type $S_i^m$ measures, with other developments like Shen's $a_i^m$, to account for competition/congestion. The more recent spatial availability measure $V_i^m$ has in common with these accessibility indicators that it is a weighted sum of the opportunities in a region from the perspective of a determined origin $i$. Aggregations of opportunities embody principles of gravitational/spatial interaction modelling that date back to at least H.C. Carey [62], and are part of a line of research that includes the work of Ravenstein [63], Reilly [64], Stewart [65–67], Zipf [68, 69], Wilson [49], and many others. In this way, $S_i^m$, $a_i^m$, and $V_i^m$ can be interpreted as scores of the potential for interaction with opportunities in space.

Different accessibility indicators are characterized by how they weight and aggregate opportunities. Spatial availability's contribution to the literature is to incorporate a proportional allocation mechanism that essentially constrains the sums to match the number of opportunities in the region: it is a singly-constrained accessibility measure that natively accommodates congestion and competition. The effort with spatial availability is in line with previous research on proportional allocation by Páez et al. (2019) [12]. As initially introduced by Soukhov et al. (2023) [15], spatial availability was designed for a homogeneous population traveling by a

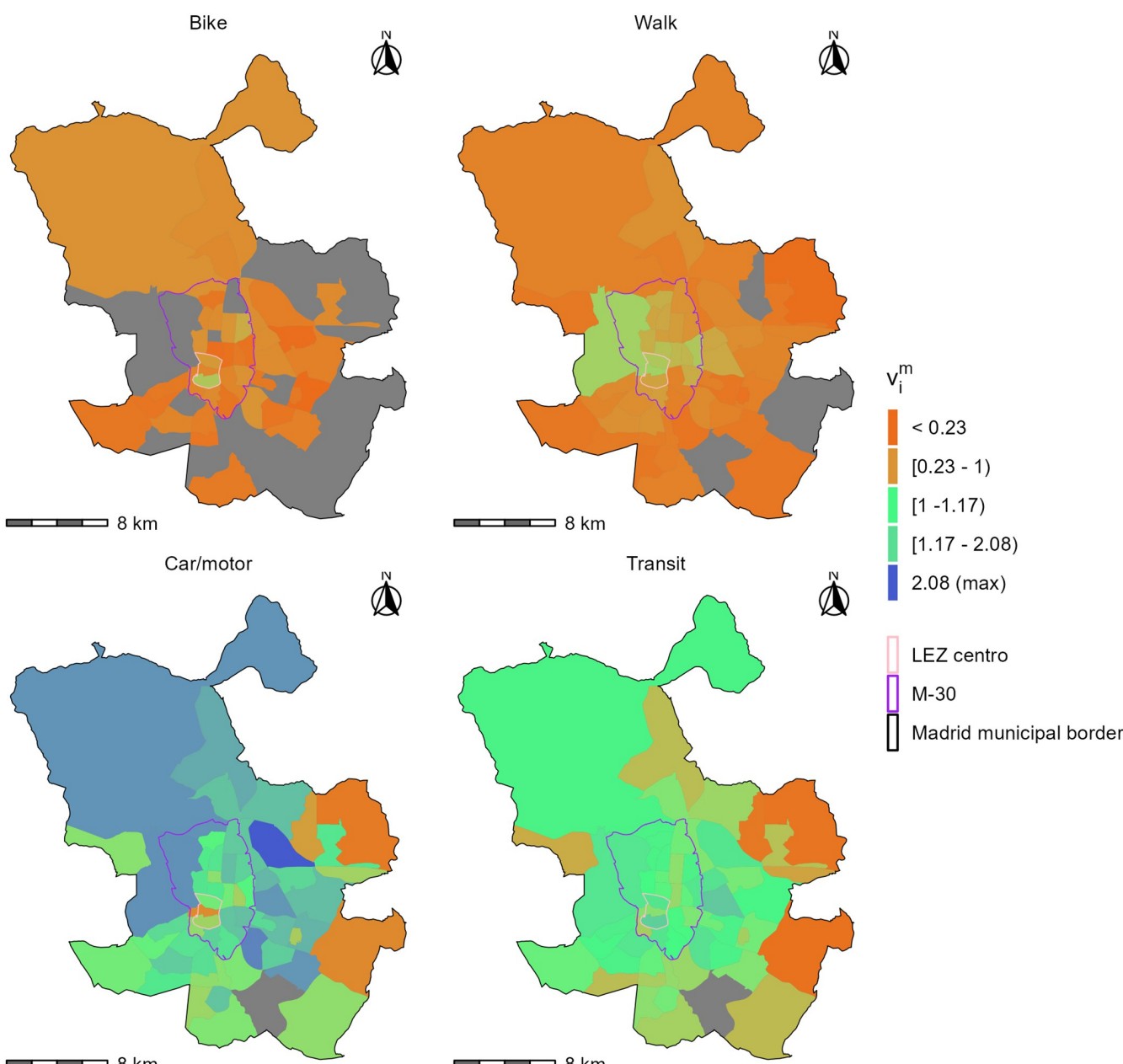

**Fig 7. Distribution of spatially available jobs per capita by mode of transportatin ($v_i^m$) represented at the level of TAZ.** Grey TAZ have no population that use the mode. Ranges of values in the legend are quintiles. The TAZ shapefile is available from the Community of Madrid open data portal.

single mode of transportation. In this paper, we extended spatial availability for the case of heterogeneous populations. We discussed this in terms of multiple modes of transportation, but the framework can accommodate equally well variations in travel behavior by population segments.

An empirical example using data from Madrid helps to illustrate the potential of multimodal spatial availability analysis, including its ability to account for competition for opportunities within and between modes. Particularly relevant is the fact that spatial availability scores relate directly to the total number of opportunities in the region. This makes it possible to

compare the results to intuitive benchmarks, such as opportunities per population, in ways that other accessibility measures cannot or tend to obfuscate. This comparability is preserved between regions and over time. The example suggests that once opportunities are treated as finite, it could be suggested that restrictions to travel by car (due to the LEZ) leave more spatially available opportunities for non-car users. This difference for car travel in locations within/around the LEZ Centro seem to increase the number of opportunities spatially available to transit users (transit being the second most competitive mode) as well as non-motorised modes. In effect, a policy such as LEZ appears to help improve the spatial availability situation of active travel and transit mode users in the parts of the city where it is implemented, though additional research is needed. To further speculate, the spatial availability allocated to car-users near but outside the LEZ Centro is still relatively high, potentially supporting the case for LEZ expansion from this perspective. The purpose of the empirical example is to illustrate the kind of insights that can be derived from the application of multimodal spatial availability.

There are some intriguing opportunities for future research. Accessibility indicators are not designed to work as modal split models, and yet, in the case of policies that alter the relative cost of various forms of transportation, one can reasonably expect to see some shifts between modes. In our empirical example, we used data collected after the introduction of LEZ Centro. However, given a modal split model to predict model shares, accessibility indicators, including spatial availability, can be used to investigate changes to the accessibility landscape. A similar logic can be applied for destination choice. Our empirical example presented a snapshot of this, and in future research it may be interesting to investigate changes in spatial availability *between* policy interventions. The plan to expand Madrid's LEZ to the ring contained by the M-30 presents an excellent opportunity. Given the intuitive interpretation of spatial availability scores as fractions of opportunities from the total, relative and absolute changes in the spatial availability landscape can be assessed, thus helping to evaluate the implications of policy interventions.

Our empirical example dealt with differences in travel by mode only, but it is possible to think of the intersection between mode of travel and different types of travelers. This would expand the number of sub-populations in the analysis from, say, $m = M$ (modes) to $m = M \cdot Q$ (modes times population segments), each with their own characteristic impedance function. Evaluations of this kind will be especially relevant as LEZ are implemented in cities globally, and the question of their impact on disadvantaged populations who have become mobility-restricted increasingly come to the fore [52, 70, 71].

To close, in this work spatial availability considers competition by allocating opportunities (the subject of the single-constraint) to modal populations in zones based on zonal proportions. Opportunity congestion can be seen on a continuum, but how it is to be considered within accessibility research is an ongoing subject of exploration [14, 17, 18]. It is in this context we present spatial availability and its multimodal extension. While our focus was on a multiple modes, this consideration is just one case of heterogeneous populations (i.e., travel by different modes). The multimodal method itself can easily accommodate other forms of population or opportunity heterogeneity, for example: variations in rich and poor, young and old, types of opportunities. Similar to preceding accessibility measures, the applications of spatial availability are as numerous as potential study contexts. To encourage open and reproducible science, the manuscript and all associated analysis is available within the lead author's GitHub repository.

## Author Contributions

**Conceptualization:** Javier Tarriño-Ortiz, Julio A. Soria-Lara, Antonio Páez.

**Data curation:** Anastasia Soukhov, Javier Tarriño-Ortiz.

**Formal analysis:** Anastasia Soukhov, Antonio Páez.

**Funding acquisition:** Anastasia Soukhov, Antonio Páez.

**Investigation:** Anastasia Soukhov, Javier Tarriño-Ortiz, Julio A. Soria-Lara, Antonio Páez.

**Methodology:** Anastasia Soukhov, Julio A. Soria-Lara, Antonio Páez.

**Project administration:** Anastasia Soukhov.

**Resources:** Anastasia Soukhov.

**Software:** Anastasia Soukhov.

**Supervision:** Anastasia Soukhov, Julio A. Soria-Lara.

**Validation:** Anastasia Soukhov.

**Visualization:** Anastasia Soukhov, Antonio Páez.

**Writing – original draft:** Anastasia Soukhov.

**Writing – review & editing:** Anastasia Soukhov, Javier Tarriño-Ortiz, Julio A. Soria-Lara, Antonio Páez.

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
