## [Decision Letter · Decision Letter 0]

11 Oct 2023

PONE-D-23-27305Multimodal spatial availability: a singly-constrained measure of competitive accessibility considering multiple modesPLOS ONE

Dear Dr. Soukhov,

Thank you for submitting your manuscript to PLOS ONE. After careful consideration, we feel that it has merit but does not fully meet PLOS ONE’s publication criteria as it currently stands. Therefore, we invite you to submit a revised version of the manuscript that addresses the points raised during the review process.

We look forward to receiving your revised manuscript.

Kind regards,

Qing-Chang Lu

Academic Editor

PLOS ONE

Journal Requirements:

"This research was funded by the Canada Graduate Scholarship - Doctoral Program (CGS D) provided by the Social Sciences and Humanities Research Council (SSHRC) and Project Mobilizing Justice, also supported by SSHRC. The funders had no role in study design, data collection and analysis, decision to publish, or preparation of the manuscript."

"AS - Canada Graduate Scholarship - Doctoral Program (CGS D) provided by the Social Sciences and Humanities Research Council (SSHRC)

AS - Project Mobilizing Justice, also supported by SSHRC.

5. We note that Figures 2, 3, 5 and 7 in your submission contain map images which may be copyrighted. All PLOS content is published under the Creative Commons Attribution License (CC BY 4.0), which means that the manuscript, images, and Supporting Information files will be freely available online, and any third party is permitted to access, download, copy, distribute, and use these materials in any way, even commercially, with proper attribution. For these reasons, we cannot publish previously copyrighted maps or satellite images created using proprietary data, such as Google software (Google Maps, Street View, and Earth). For more information, see our copyright guidelines: http://journals.plos.org/plosone/s/licenses-and-copyright.

1.) You may seek permission from the original copyright holder of Figures 2, 3, 5 and 7 to publish the content specifically under the CC BY 4.0 license.  

2.) If you are unable to obtain permission from the original copyright holder to publish these figures under the CC BY 4.0 license or if the copyright holder’s requirements are incompatible with the CC BY 4.0 license, please either i) remove the figure or ii) supply a replacement figure that complies with the CC BY 4.0 license. Please check copyright information on all replacement figures and update the figure caption with source information. If applicable, please specify in the figure caption text when a figure is similar but not identical to the original image and is therefore for illustrative purposes only.

Reviewers' comments:

Reviewer's Responses to Questions

**Comments to the Author**

1. Is the manuscript technically sound, and do the data support the conclusions?

Reviewer #1: Partly

Reviewer #2: Yes

Reviewer #3: Yes

2. Has the statistical analysis been performed appropriately and rigorously? 

Reviewer #1: Yes

Reviewer #2: Yes

Reviewer #3: Yes

3. Have the authors made all data underlying the findings in their manuscript fully available?

Reviewer #1: Yes

Reviewer #2: No

Reviewer #3: Yes

4. Is the manuscript presented in an intelligible fashion and written in standard English?

Reviewer #1: Yes

Reviewer #2: Yes

Reviewer #3: Yes

5. Review Comments to the Author

Reviewer #1: The article extends the concept of spatial accessibility and applies it to a fairness analysis of different modes of transportation. The article uses travel data from a week in Madrid as an example to analyze spatial accessibility in the city. However, the article has the following issues:

1. Lack of Innovation: The article's improvement on spatial accessibility mainly involves incorporating the ratio of travel frequencies as weight in travel costs. This improvement is relatively minor and the method used does not demonstrate superiority over traditional accessibility approaches.

2. Insufficient Experimental Data: The article uses questionnaire data from a week in Madrid, but lacks basic descriptions of the questionnaire data. Additionally, the daily travel data of approximately 30,000 trips is significantly limited, and there is no description of the criteria for selecting the dates. The typicality of the experimental data is questionable.

3. Lack of Data-Driven Analysis: In the analysis section, a large amount of data is used to analyze people's travel behavior in different areas. The analysis of spatial accessibility only considers the differences in the modes of travel mentioned earlier. The analysis results are somewhat one-sided, and examining the indicator from multiple perspectives would provide a more objective view. It is recommended to expand the dimensions of the analysis.

4. Disorganized Format: The article's methodology section contains numerous formulas and variables, but the definitions of these variables are unclear and difficult to read. There are also numerous formatting errors, and Table 1 is disorganized and unappealing. It is recommended to revise these issues.

Reviewer #2: This paper has sound mathematical foundations and allows to answer its research question in an elegant way (how to measure competition for e.g. jobs based on spatial accessibility?). I also really appreciated the fact that the paper was very didactic. Still, I have concerns about the relevance of the paper for future research.

More specifically, the new measure proposed by the authors has clear limitations:

i) it focuses on the competition for jobs and is not useful for studying access to non-competitive or semi-competitive resources such as amenities.

ii) it doesn't allow to study absolute gains or losses in accessibility from public transportation infrastructure improvements or changes.

iii) the authors do not allow for modal shifts: they assume that the transport mode choice of households is fixed and cannot evolve due to e.g. transport infrastructure changes.

These points limit the relevance of the new accessibility measure. The authors should, at least, specify these limitations early in the paper. The introduction should start by stating the precise research question the new accessibility measure is seeking to answer as well as its limitations). They should also justify, based on the literature, that competition for jobs is a key determinant of job market outcomes, and that there is strong inertia in mode choices.

Finally, the writing of the paper, and particularly of the abstract and introduction, should be improved. The abstract could state the broader relevance of the topic and summarize the results of the case study on LEZs. The introduction should start more directly by introducing the research question and its relevance and describing the new measure and its limitations.

Minor comments:

i) what are the summary statistics p10 (car: 36 min, transit: 55 min,...)? I assume they correspond to the mean.

ii) I also have identified a few formatting issues (e.g. Fcij p10 and 4.72km2 p 8).

Reviewer #3: This manuscript extended the authors’ previous work spatial availability measure, which is a type of location-based accessibility measure that is both constrained and competitive compared to Hansen-type measure and Shen-type accessibility measure, into a multimodal framework. The new measure, multimodal spatial availability, strengthened the constrained (or finite) nature of opportunities, and the competitive nature among multimodal accessibility resulting from this constraint through a synthetic example and an empirical example of the LEZ in the city of Madrid. In conclusion, the authors demonstrated one restriction had impacted the spatial availability of opportunities for other modes using and proposed potential future uses in policy planning scenarios.

In general, the manuscript was logical and well-structured. The research problem was well defined. The data were available and quite supported the conclusion. The statistical analysis performed appropriately.

However, there are some issues:

Major issues:

Please demonstrate whether “car/motor & transit” and “bike & walk” are comparable or whether they are in an actual competitive relationship? For example, if I work 3km from where I live, maybe I will never choose to take a transit, I will always walk or ride. But if I work 20km from where I live, walking or riding to work seems impossible for me, I have to drive or take public transportation. Car/motor and transit can be in competitive relationship and people can choose which one they prefer, but not choose between motor and walk. This issue will also have an impact on the results of the research.

Minor issues:

1. As for Fig 2 and Fig 3, please indicate the meaning of the gray color blocks in illustration.

2. Please change the color scheme of fig 2. The red color scheme makes the LEZ centro area boundary, which is also in red, not visible.

6. PLOS authors have the option to publish the peer review history of their article (what does this mean?). If published, this will include your full peer review and any attached files.

Reviewer #1: No

Reviewer #2: **Yes: **Charlotte Liotta

Reviewer #3: **Yes: **Xin Xu

---

## [Author Response · Author response to Decision Letter 0]

7 Dec 2023

Please see the 'Response-to-reviewer.pdf' for the formatted letter.

#Reviewer 1

Thank you for this comment regarding "1. Lack of Innovation". We would like to bring to your attention the editorial philosophy of PLoS ONE, according to which editors "make decisions on submissions based on scientific rigor, regardless of novelty," (see https://journals.plos.org/plosone/s/editorial-and-peer-review-process). Nonetheless we wish to respond to this comment.

In this paper we extend our earlier work on spatial availability (https://doi.org/10.1371/journal.pone.0278468) for the simultaneous analysis of multiple modes of transportation. You appear to be under the misaprehension that the main difference with accessibility measures is to incorporate "the ratio of travel frequencies as weight in travel costs". This statement is not correct. The main difference is the proportional allocation mechanism that we introduced for spatial availability, and that is extended in this paper to allocate opportunities based on the proportion of travelers by different modes. This is not the same as "weighting the travel cost". In fact, each mode is modelled using its own impedance function, as shown in our empirical example, where we use origin-destination data by mode to estimate travel impedance functions specific to each mode.

While this enhancement may appear minor in your opinion, it is not plainly obvious how spatial availability applies to multiple modes, which is why we believe this paper is needed. We do not dispute that this is an incremental step in the development of a more general method, but it is a step that considerably expands the

range of potential applications. Further, we contend that the method has been rigorously developed and demonstrated this using an open, transparent, and reproducible example. In response to this comment we have edited the paper to more clearly describe the advantages that multimodal spatial availability offers when considering the accessibility to opportunities by different modes. In particular, proportional allocation of opportunities using mode-specific impedance functions means that the travel- cost-advantage of each mode can be analysed. Further, this mechanism ensures that the sum of all spatial availability values for all modes sum up to the total number of opportunities in the region, which is not true for any other type of accessibility measure. With respect to the "superiority" of the method, a reader of our earlier paper would already be aware of what limitations of accessibility analysis spatial availability aims to address. In this paper we also try to demonstrate throughout the manuscript what our measure does that others can’t. Since spatial availability values result from proportional allocation, each value is a proportion of 

the total number of opportunities. Put another way, we can compare the proportion of opportunities available to users of each mode, to users at each zone, to users of each mode by zone, and so on, and the values relate directly to the total opportunities in the region. This also allows us to calculate values per capita that serve as benchmark values, as shown in the empirical example.

Thank you for this comment regarding "2. Insufficient Experimental Data". To clarify, the empirical data used is not experimental. It is observational, since it is collected using a travel survey conducted by the City of Madrid. Travel surveys are a standard instrument in transportation planning and research, and are conducted in cities around the world. The data we work with represent the most recent and most complete travel survey conducted for the region to date. We are not completely sure where you got the figure of "30,000 trips" (which you consider significantly limited); presumably you are citing the maximum ’opportunity’ or ’population’ numbers in Figures 2 or 3. The most job-rich TAZs have 30,000 jobs while the least have 1,000 or fewer (Figure 2). In fact, there are 847,574 jobs in the city, which is also the sum of the total spatial availability in our analysis, as well as the number of potential trips to work. 

With respect to The "typicality" of the data, travel surveys are designed to provide information about personal travel on a typical day, usually during a period of maximum demand (i.e., not during the summer vacation). This is standard practice for these surveys, and the one for Madrid is no different in this respect.

Regarding "3. Lack of Data-Driven Analysis". Thank you for this comment. It is somewhat puzzling that "30,000 trips" would be significantly limited (as per your comment #2), and at the same time be "a large amount of data". We find that this comment in particular is not actionable due to its vagueness. There are no meaningful responses to "somewhat one-sided" when "somewhat" and "one-sided" do not quantify or refer to a particular side. We studied the modes available

in the region, and examined the results from the perspective of each mode. What other perspectives do you suggest? What dimensions should be explored?

Despite this comment being nonactionable, we believe that with the additional detail added to the manuscript to address your and other reviewers’ comments, that overall clarity has been improved.

Regarding "4. Disorganized format". Reviewer #2 identified some specific formatting issues and we fixed them. Otherwise we proofread the paper and hopefully did not leave a typo behind. We also made sure Table 1 starts on a new page, when it runs across pages the formatting resulted in some disorganization. Thank you again for your efforts reviewing our submission.

---

#Reviewer 2

Thank you for your thoughtful review of our paper. Your comments were very helpful to improve the clarity of the research.

Regarding "method limits: focuses on the competition for jobs and is not useful for studying access to non-competitive or semi-competitive resources". Thank you for this comment. We would begin by noting that there are many types of opportunities that are mutually exclusive due to competition. For this paper we focused on jobs, but there are many others, such as beds at hospitals, seats at schools, and so on. Thus, even if the measure was applicable only to exclusive opportunities, there are numerous applications to choose from.

That said, we have grappled for some time with the question of what types of opportunities are best analysed using spatial availability. Our current thinking on this matter, after much consideration, is that in practice every type of opportunity, even when not clearly exclusive, is subject at least to congestion or capacity constraint. For instance, green spaces are often considered non-competitive, however, standards for the provision of such amenties are provided in the form of units of amenity per capita. For example, a case could the Ile-de-France region, a jurisdiction that suggested in a major planning document of 2013 that at the municipal level at least 10m^2 of public green space should be supplied per inhabitant (Liotta et al. 2020). But green spaces are not evenly distributed, which means that who has access to them hinges on where they are and how easy is to reach them. Formulating the provision of amenities in these terms in not rare. For example, Natural England recommends an Accessible Natural Greenspace Standard such that the minimum supply of space is one ha of statutory Local Nature Reserves per thousand population (see https://redfrogforum.org/wp-content/uploads/2019/11/67-Nature-Nearby%E2%80%99-Accessible-Natural-Greenspace-

Guidance.pdf) . Similarly, the World Health Organization (cited in OECD, 2013) recommends that cities provide a minimum of 9m^2 of green area per inhabitant (see https://doi.org/10.1787/9789264191808-en). For our purposes, standards of this type translate into "how much of this resource is available to one individual that has not been claimed by anyone else?". Green spaces often have large capacities, but they still have a capacity, and it is not the same for a person to have access to 5m^2 of uncongested green space than to 15m^2. This difference is in fact a matter of justice (Lara-Valencia and García-Pérez 2015; Liotta et al. 2020). Constraining accessibility is in this way a useful way to evaluate the congested availability of any type of opportunity. As standards are emphasized in the planning literature, in particular for fairness in transportation (see Martens and Golub 2021), spatial availability analysis can be used to assess standards. We are convinced that as other researchers discover this new approach to measuring accessibility many other applications will be found.

Regarding "method limits: does not allow for absolute gains or losses to be studied". Thank you for this comment. Spatial availability can certainly be used to capture absolute gains and losses. In fact, logically, the gains and losses produced by using a competitive and constrained measure allows for a clear interpretation. This would require the analyst to estimate the accessibility before and after some change to the land use or transportation system. In this paper our empirical application is a single scenario to serve as proof of concept, but in future research we intend to use spatial availability to analyse changes in the size of Madrid’s Low Emissions Zone implementation from a modal and socio-economic equity perspective.

Regarding "method limits: modal shift?". This is an excellent point. Accessibility measures, including spatial availability, are not meant to function as modal split models, but they are certainly amenable to analysis of changes in the accessibiity landscape if different modal shares are used as inputs. This is similar to the case of destination choice: accessibility measures do not model this, but changes in destination choices can be incorporated via how they affect the impedance function. In terms of how this could be implemented, the process can be sketched as follows: the results of a modal split model are used to estimate new modal shares, which in turn are used to recalculate spatial availability. The new values then can be compared to the baseline scenario. In other words, the framework for spatial availability is sufficiently flexible to take in not only mode-specific travel impedance functions, but also the proportions of the populations using each mode.

Regarding "suggestions for specifying method limits". Your comments have been very helpful to improve the clarity of the paper, as well as the scope of what we do, as well as directions for future research. For example, we now note in the "Discussion and conclusions" that "our example dealt with differences in travel by mode only, but it is possible to think of the intersection between mode of travel and different types of travelers. This would expand the number of sub-populations in the analysis from, say, m = M (modes) to m = M · Q (modes times population segments), each with their own characteristic impedance function. Evaluations of this kind will be especially relevant as LEZ are implemented in cities globally, and the question of their impact on disadvantaged populations who have become mobility-restricted increasingly come to the fore (De Vrij and Vanoutrive 2022; Verbeek and Hincks 2022; Liotta 2023)."

Regarding "improvements to abstract and introduction". We have now done this. In the original version of the abstract and introduction we gave the misleading impression that we would analyse changes in the system, when in reality our intent was to demonstrate the application of the measure in an empirical example. We do plan to study changes in the system, but this requires more work than can be presented in a single paper, partly for the reasons that you identified above (modal shifts and the calculation of spatial availability for two different scenarios, as well as the analysis of the differences between them). We plan to do this in a future paper focused on the policy instead of the presentation of a new method. Thank you again for your thoughtful comments and suggestions to improve the paper.

Regarding "minor comments". Updated! Apologies, car: ’36 min’ corresponds to a mean of 36 minutes and then within the brackets

additional descriptive statistics (minimum value, maximum value, etc.). We also fixed the formatting issues specified.

---

#Reviewer 3

Regarding "major issues". This is an excellent comment. The short answer is that whenever a destination can be reached by more than one mode, users of those modes are in competition for the opportunities there.

In this revision we tried to improve the discussion to make this point more clear. The impedance functions for all four modes are not the same. They describe the travel behaviour of commuters as informed by the 2018 travel survey. To follow your example, someone who cannot walk to work because their job is 20 km from where they live, will not compete for that job against people from their same origin who do walk. However, if the place where their work is can be reached by anyone who can walk from other origins (say, someone who lives closer to that destination), they would be in competition for the same opportunity.

Furthermore, average travel times for car/motor and transit are longer than bike and walk. All people don’t have access to all options - completely true. But the travel impedances reflect this real travel at an aggregate based on all the trips for a mode. And on average, it is assumed that people at each origin that take a mode to a destination are in direct competition for opportunities (as opportunities are finite) – and a part of the competition is defined by the mode-specific impedance function (the second part is the population balancing factor). This assumption, that all populations, no matter their mode, are competing for the same finite set of opportunity, is part of spatial availability. We’ve made an effort to make this more clear in the text.

Regarding "minor issues". We have updated the captions of the figures to reflect the meaning of grey colour blocks and the colour scheme of all Figures. We’ve also updated the labels on the legends to make them more interpretable.

---

## [Decision Letter · Decision Letter 1]

4 Jan 2024

PONE-D-23-27305R1Multimodal spatial availability: a singly-constrained measure of accessibility considering multiple modesPLOS ONE

Dear Dr. Soukhov,

Thank you for submitting your manuscript to PLOS ONE. After careful consideration, we feel that it has merit but does not fully meet PLOS ONE’s publication criteria as it currently stands. Therefore, we invite you to submit a revised version of the manuscript that addresses the points raised during the review process.

Please respond to the reviewers' comments carefully and address them one by one. 

We look forward to receiving your revised manuscript.

Kind regards,

Qing-Chang Lu

Academic Editor

PLOS ONE

Journal Requirements:

Additional Editor Comments:

Please respond to the reviewers' comments carefully and address them one by one.

Reviewers' comments:

Reviewer's Responses to Questions

**Comments to the Author**

1. If the authors have adequately addressed your comments raised in a previous round of review and you feel that this manuscript is now acceptable for publication, you may indicate that here to bypass the “Comments to the Author” section, enter your conflict of interest statement in the “Confidential to Editor” section, and submit your "Accept" recommendation.

Reviewer #2: (No Response)

Reviewer #3: All comments have been addressed

2. Is the manuscript technically sound, and do the data support the conclusions?

Reviewer #2: Yes

Reviewer #3: Yes

3. Has the statistical analysis been performed appropriately and rigorously? 

Reviewer #2: Yes

Reviewer #3: Yes

4. Have the authors made all data underlying the findings in their manuscript fully available?

Reviewer #2: Yes

Reviewer #3: Yes

5. Is the manuscript presented in an intelligible fashion and written in standard English?

Reviewer #2: Yes

Reviewer #3: Yes

6. Review Comments to the Author

Reviewer #2: I appreciate the fact that the authors have largely revised the paper to answer the reviewers’ comments. Still, I feel like some of my comments have been insufficiently addressed in the revised draft, as detailed below.

The authors have adequately addressed my comment on the fact that their new measure is not useful for studying access to non-competitive resources in the revised introduction.

My comment on the fact that the new measure doesn’t allow to study the absolute gains or losses in accessibility from public transport or land use system changes has inadequately been accounted for. First, what your new measure can or cannot do remains unclear to me: for instance, how would an improvement in the transport sector that benefits all transport modes and locations uniformly, but still reduces all transportation times in the city, be accounted for? Second, I do not see where this point is discussed in the paper.

My comment on modal shift has partially been accounted for. A key question is whether people can shift transport modes easily or not: indeed, it seems to be that your new accessibility measure, based on competition, does only make sense in case people cannot easily shift transport modes. You partially answer that question in the paragraph starting l46 by explaining why some populations, e.g. children, elderly, or single parents, might have different characteristics in terms of transportation choice and mode use. Still, in the case of the Madrid LEZ, you do not explain why private cars and public transport users are in competition. For instance, you state that “restrictions to travel by car leave more spatially available opportunities for non-car-users” (l585). But it seems to me that the LEZ aims at promoting a modal shift toward less polluting cars, public transport, or active modes rather than favoring public transport users? Relatedly, your measure does not make any difference between car users that can easily shift to public transport, and car users that cannot.

My other comments have been accounted for, and the writing of the paper has largely been improved.

Finally, I have a few additional minor comments:

Fig 5: the boundaries of the LEZs, and in particular of the LEZ centro, are really hard to see.

There is a typo l56: “The paper rest of the paper”

Reviewer #3: (No Response)

7. PLOS authors have the option to publish the peer review history of their article (what does this mean?). If published, this will include your full peer review and any attached files.

Reviewer #2: **Yes: **Charlotte Liotta

Reviewer #3: **Yes: **Xin Xu

---

## [Author Response · Author response to Decision Letter 1]

1 Feb 2024

Please see the 'Response-to-reviewer.pdf' for the formatted letter.

# Reviewer 1

NA

---

# Reviewer 3

NA

---

# Reviewer 2

Thank you for for the time and care taken by the editorial team and reviewers in providing feedback to the second round of revision to the manuscript. No doubt, the further revised version submitted is more articulate than the first. Our responses to comments to Reviewer #2 are shown below in \\textcolor{blue}{blue}. Reviewer #3 and #1 did not have any additional comments.

## Reviewer #2: 

I appreciate the fact that the authors have largely revised the paper to answer the reviewers’ comments. Still, I feel like some of my comments have been insufficiently addressed in the revised draft, as detailed below.

\\textcolor{blue}{Thank you: comments provided by all reviewers were exceptionally helpful in clarifying the manuscript. We address the outstanding three comments and additional minor comments in this response letter.}

The authors have adequately addressed my comment on the fact that their new measure is not useful for studying access to non-competitive resources in the revised introduction.

\\textcolor{blue}{Thank you, we appreciate your understanding.}

My comment on the fact that the new measure doesn’t allow to study the absolute gains or losses in accessibility from public transport or land use system changes has inadequately been accounted for. First, what your new measure can or cannot do remains unclear to me: for instance, how would an improvement in the transport sector that benefits all transport modes and locations uniformly, but still reduces all transportation times in the city, be accounted for? Second, I do not see where this point is discussed in the paper.

\\textcolor{blue}{This is a great question, and we think we know the answer. We interpreted your comment initially as to mean changes in accessibility, and we responded accordingly. Your example makes it clear that you had something else in mind. The answer to your question can be found in point 2 in the conclusion of Soukhov et al.} \\begingroup\\color{blue}[-@soukhovIntroducingSpatialAvailability2023]\\endgroup \\hspace{1em}\\textcolor{blue}{ (p. 25) where it says "Shen-type accessibility can be used to compute the availability of jobs (the rate and the absolute values if the original definition is corrected), however, if the analyst is interested in internal values and secondary analysis of the results, spatial availability should be considered". The internal values referred to there are the estimates of trip length/duration/cost associated with the spatial availability of opportunities; the secondary analysis, in turn, referred to estimating the system-wide cost of competitive accessibility. To use your example, suppose that there is a change in the transportation system that is uniform: accessibility does not change but travel times are shorter for every mode. As Soukhov et al.} \\begingroup\\color{blue}[-@soukhovIntroducingSpatialAvailability2023]\\endgroup \\hspace{1em}\\textcolor{blue}{demonstrate in the section \\textit{Why does proportional allocation matter?} (p. 15), a difference between spatial availability and unconstrained accessibility measures, is that spatial availability allocates the actual number of opportunities to each origin. As a result, the estimates of travel time can be used to quantify the system-wide impacts of spatial availability. In your hypothetical example, spatial availability would remain constant, but the system-wide cost of travel, as calculated from the intermediate values (i.e., the detailed table of opportunities allocated to origins by origin-destination pair), would decrease.}

My comment on modal shift has partially been accounted for. A key question is whether people can shift transport modes easily or not: indeed, it seems to be that your new accessibility measure, based on competition, does only make sense in case people cannot easily shift transport modes. 

\\textcolor{blue}{Thank you for your comment. For the empirical example we use data collected at least one year \\textit{after} the introduction of Madrid LEZ. We then make the (hopefully reasonable) assumption that all modal shifts caused by LEZ have already taken place. We could (if we had the data) calculate spatial availability using the mode shares \\textit{before} the introduction of Madrid LEZ. Here, however, we are not conducting a before/after comparison of the policy, but only of the spatial availability already in the context of LEZ. In other words, our discussion is based not on a comparison over time, but a comparison over space. We have further enhanced the results to make this interpretation more clear. }

You partially answer that question in the paragraph starting l46 by explaining why some populations, e.g. children, elderly, or single parents, might have different characteristics in terms of transportation choice and mode use. Still, in the case of the Madrid LEZ, you do not explain why private cars and public transport users are in competition. For instance, you state that “restrictions to travel by car leave more spatially available opportunities for non-car-users” (l585). 

\\textcolor{blue}{Our discussion of competition is not based on hypothetical modal shifts caused by LEZ, but rather on the comparison of spatial availability after LEZ over space. Fig. 6 in the paper shows that bike and walk consistently have fewer jobs available than their share of users in the population. But, they receive a larger share \\textit{within} the boundaries of Centro, where the number of cars that can enter is limited by the LEZ policy. In other words, the restriction, which is what distinguishes Centro from the boundary of the M-30 or the whole region, seems to open up a window of opportunity for jobs to become available to cyclists and pedestrians.}

But it seems to me that the LEZ aims at promoting a modal shift toward less polluting cars, public transport, or active modes rather than favoring public transport users? 

\\textcolor{blue}{No question that promoting a modal shift is part of the policy. However, as we noted above, we assume that the data already incorporates said shift. Given the modal shares, our analysis indicates that the winning mode within the boundaries of LEZ is transit (Fig. 6).} 

Relatedly, your measure does not make any difference between car users that can easily shift to public transport, and car users that cannot.

\\textcolor{blue}{You are absolutely right. That said, the job of allocating users to various modes is not the job of our measure, but the job of a mode split or mode choice model. The scope of our measure is to allocate opportunities to users by mode. Given changes in the shares of different modes, the measure can be calculated to examine the spatial availability under those conditions.} 

My other comments have been accounted for, and the writing of the paper has largely been improved.

Finally, I have a few additional minor comments:

Fig 5: the boundaries of the LEZs, and in particular of the LEZ centro, are really hard to see.

There is a typo l56: “The paper rest of the paper”

\\textcolor{blue}{Thank you bringing these comments to our attention. We have changed the colours and slightly increased the size of the spatial borders for LEZ Centro and M-30 on all figures (now the borders are Pink and Purple instead of Light and Medium Grey) to improve visibility. We have also corrected the identified typo. Additionally, we have improved phrasing and corrected grammatical errors throughout the manuscript. }

\\textcolor{blue}{We wish to thank you for your thoughtful and constructive comments throughout the revision process so far. A sticking point is the shift in modes. In this respect, we would like to reiterate that our measure is not intended to do the things that modal split or mode choice models are supposed to do. Our empirical example is cross-sectional, and we believe bypasses the question of modal shifts while still giving useful information about the spatial distribution of opportunities and the ability of different modes to reach them. We would argue that the materials presented here are already an advance over existing methods--even in the case when accessibility remains constant but the cost of travel declines uniformly. Still, we wish to acknowledge that the use of multimodal spatial availability in over-time comparisons would benefit from being paired with a modal split or mode choice model, something that we are prevented from trying in our empirical example due to data limitations.}

---

---

## [Editor Report · Decision Letter 2]

5 Feb 2024

Multimodal spatial availability: a singly-constrained measure of accessibility considering multiple modes

PONE-D-23-27305R2

Dear Dr. Soukhov,

We’re pleased to inform you that your manuscript has been judged scientifically suitable for publication and will be formally accepted for publication once it meets all outstanding technical requirements.

Kind regards,

Qing-Chang Lu

Academic Editor

PLOS ONE
---

## [Editor Report · Acceptance letter]

8 Feb 2024

PONE-D-23-27305R2 

PLOS ONE

Dear Dr. Soukhov, 

I'm pleased to inform you that your manuscript has been deemed suitable for publication in PLOS ONE. Congratulations! Your manuscript is now being handed over to our production team.

Kind regards, 

on behalf of

Dr. Qing-Chang Lu 

Academic Editor

PLOS ONE